# Artificial Neural Network Study on the Pyrolysis of Polypropylene with a Sensitivity Analysis

**DOI:** 10.3390/polym15030494

**Published:** 2023-01-18

**Authors:** Ibrahim Dubdub

**Affiliations:** Department of Chemical Engineering, King Faisal University, Al-Hassa 31982, Saudi Arabia; idubdub@kfu.edu.sa

**Keywords:** machine learning, pyrolysis, ANN, TGA, polypropylene

## Abstract

Among machine learning (ML) studies, artificial neural network (ANN) analysis is the most widely used technique in pyrolysis research. In this work, the pyrolysis of polypropylene (PP) polymers was established using a thermogravimetric analyzer (TGA) with five sets of heating rates (5–40 K min^−1^). TGA data was used to exploit an ANN network by achieving a feed-forward backpropagation optimization technique in order to predict the weight-left percentage. Two important ANN model input variables were identified as the heating rate (K min^−1^) and temperature (K). For the range of TGA values, a 2-10-10-1 network with two hidden layers (Logsig-Tansig) was concluded to be the best structure for predicting the weight-left percentage. The ANN demonstrated a good agreement between the experimental and calculated values, with a high correlation coefficient (R) of greater than 0.9999. The final network was then simulated with the new input data set for effective performance. In addition, a sensitivity analysis was performed to identify the uncertainties associated with the relationship between the output and input parameters. Temperature was found to be a more sensitive input parameter than the heating rate on the weight-left percentage calculation.

## 1. Introduction

Researchers are currently shifting towards a powerful artificial neural network (ANN) model analysis to handle linear and non-linear systems. Therefore, in this work, ANN will be implemented as new technique to analyze the pyrolysis reaction of PP. Muravyev et al. (2021) [1] reported that there is no mathematical tool that can solve all the pyrolysis TGA problems. In some cases, the ANN may become an alternative option.

There are two earlier-reported publications, including my own publication, for TGA data analysis using ANN [2,3,4]. In this research, Dubdub (2022) [4] summarized and presented most of the reported publication list from before the year of 2022 with different architecture details for non-isothermal TGA data.

In this introduction, all the papers that used the ANN technique for TGA data and were published within the current year (2022) have been reviewed. Usually, ANN networks agree to be encoded numerically, as “(x-y-z).” For the neurons, these x, y, and z variables denote the input layer, hidden layer, and output layer (Muravyev et al. (2021) [1]), respectively. Sometimes, more than three numbers exist, such as “(x-y-z-a),” which means there are two hidden layers with y, z neurons. This information is mentioned in the early stages of this paper because some of these codes will be used in the introduction section.

Khan and Taqvi (2022) [5] reviewed the applicability and various benefits of machine learning methods such as in prediction, fault detection, optimization, and quality control. They classified ML into two main types: supervised learning techniques, including ANN, and unsupervised learning techniques. The authors also mentioned different applications of ANN in Chemical Engineering process systems, including chemical reaction processes, the auto tuning of PI and PID controllers, coagulation in poly aluminum chloride, water-treatment plants, the optimization and modeling of cross-flow ultrafiltration, the performance of wastewater treatment processes, oil degradation in wastewater, water quality in terms of chemical oxygen demand (COD) and biological oxygen demand BOD [6], and the removal of zinc ions from wastewater in hydrodynamic cavitation for biomass pretreatment.

Gonzalez-Aguilar et al. (2022) [7] reported the effects of heating rate and temperature on the thermal pyrolysis of expanded polystyrene with respect to liquid conversion yields and physicochemical properties in a semi-batch reactor using factorial statistical analysis.

Demir (2022) [8] used different compositions of polycaprolactone (PCL)/polyvinyl chloride (PVC) blends in order to study the thermal degradation kinetics with a four-heating rate using ANN analysis. Further, the investigated results of the decomposition temperature, heating rate, and percentage of PVC in blends as input data for the ANN model, and the percentage of weight remaining during degradation as output data for the ANN model were reported by [8]. He concluded the 3-10-10-1 network with Logsig–Tansig transfer function with feed-forward backpropagation was the best network for this system. He then developed another system with a 4-10-10-1 topology with a Logsig–Logsig transfer function by taking the degradation temperature, heating rate, percentage PVC, and percentage weight left data as input data, and using the activation energy values as output data.

Ai et al. (2022) [9] investigated the use of ANN for the co-pyrolysis of oily sludge and high-density polyethylene (HDPE). They used two models: one that predicted the interactive effect of HDPE using the heating rate, temperature, and mixing ratio as the input variables, and a second model that predicted the activation energy with the mixing ratio and reaction conversion degree as the input variables. The performance and validation of the model was determined by the average regression coefficient (R^2^) and by measuring the root mean squared error (RMSE) calculation. For both cases, authors used two hidden layers with (8–10) neurons (R^2^ = 0.99) and (10–7) neurons (R^2^ = 0.92) for the first and second models, respectively.

Balsora et al. (2022) [10] studied the pyrolysis of lignocellulosic biomass by an ANN. They used the preliminary analyses, including proximate, ultimate, and biochemical analysis, as input variables (four different input set variables) and the kinetic parameters as output variables by four different models of an ANN network. They used the mean square error (MSE) statistical-criteria parameter to obtain the optimum number of neurons in the hidden layer and found 9, 6, 200, and 39 neurons for ANN-1, ANN-2, ANN-3, and ANN-4, respectively. They tried to improve the performance of the first two ANN models by combing the number of input variables to obtain the third ANN model, and they improved further in the fourth ANN model by including the biochemical analysis as additional input for the input variables of the third ANN model. As a sensitivity analysis study, they highlighted that the effect of the biomass analysis on the output variables estimation was very significant. Jacob et al. (2022) [11] studied the ANN to predict the mass loss of a raw-mustard biomass using three hidden layers and one output layer. They used the mixing ratio, heating rate, and temperature as input variables and the mass loss as only output variable.

Kartal and Özveren et al. (2022) [12] applied the ANN to the pyrolysis of almond shell and imbat coal. They focused their work on estimating the activation energy for the two samples by using the ultimate analysis and the experimental TGA data for the ANN model. These inputs include the carbon content, oxygen content, hydrogen content, temperature onset, temperature end, heating rate, flowrate of gas, type of thermochemical process, and particle size ranges. They used, out of total 188 datasets, 80% data for training, 5% for validation, and the remaining 15% for testing. Furthermore, they used a Tansig activation function for the two hidden layers with (12-6) neurons in the first and second hidden layers, respectively.

Khodaparasti et al. (2022) [13] used (2-8-1) as network architecture for an ANN network, with the temperature and heating rate as input variables and the remaining mass (%) as the target or output variable for the co-pyrolysis of municipal sewage sludge and microalgae chlorella vulgaris. They found that the R^2^ value for this network was close to one, indicating that the ANN prediction was very close to the experiential data. 

Li et al. (2022) [14] applied the ANN (multi-layer, feed-forward, network-backpropagation) for one run with a heating rate 20 K min^−1^ for the co-pyrolysis of agricultural waste and HDPE. They applied the mass change of feedstock as the output variable and temperature as the input variable. They concluded the best architecture was three neurons in one hidden layer with a sigmoid function.

Nawaz and Kumar et al. (2022) [15] studied the ANN for the pyrolysis of a biomass (Sesbania bispinosa). They found that the best structure was (2-10-1) by considering R^2^ and MSE as the main criteria. Similar to many other papers, they used the temperature and heating rate as input variables and the weight-loss percentage as the output variable.

Postawa et al. (2022) [16] used the ANN differently when compared to the above-mentioned researchers by selecting four parameters: the heating rate, activation energy, pre-exponential factor, and reaction order as input variables and choosing the percentage shares of cellulose, hemicellulose, and lignin as output variables. The result was a (4-11-16-3) structure in which two hidden layers of 11 and 16 neurons were considered to be the first and second hidden layers, respectively. 

Dubdub and Al-Yaari (2022) [4] and Dubdub and Al-Yaari (2021) [17] employed the same ANN technique to check the validity for TGA data. They used a feed-forward backpropagation ANN model with two hidden layers to predict the TGA data. In the first cited paper [4], they applied two input variables: the heating rate (K min^−1^) and the temperature (K); in the second paper, the ratio between the catalyst and the polymer was included as a third input [17]. 

In this present paper, a highly efficient, developed ANN model has portended the pyrolysis behavior of PP polymer using TGA data. Moreover, a sensitivity analysis has been conducted to find the uncertainty in the relationship between the input variables and the output variables. Finally, temperature has been found to be a more sensitive input parameter when compared to the heating rate variable.

## 2. Materials and Methods

### 2.1. The Experiments

TGA experiments were conducted in nitrogen at 100 (cm^3^ min^−1^) with five heating rates, as is shown in Figure 1 and Figure 2, respectively. More details on the TGA brand (PerkinElmer Co., Waltham, MA, USA) and the procedure, including material characteristics such as proximate and ultimate analysis data, are available in my latest publication [18].

### 2.2. ANNs Topology

The modelling of any process is not easy task to begin with, and the values of the model parameters are usually evaluated from process data using advanced numerical techniques. In spite of developing a model and considering the assumptions, these parameters may yet require difficult tasks in some cases to solve. In addition, there is no mathematical tool that can solve all pyrolysis–TGA problems (Muravyev et al. (2021) [1]). This becomes increasingly difficult with a non-linear system. Therefore, ANN may be the efficient tool for providing the best solution.

ANN architecture is mainly managed in a succession of three layers, with every layer having a bias, a weight, and an output (Quantrille and Liu (1992) [19]). Initially, we should consider that any variables having an effect on the process should be resolved before utilizing the ANN network. Data collection must represent the problem, with the input and output variables and all the data placed within the range for each variable. 

The performance of an ANN architecture should be tested by varying the number of hidden layers, the transfer function of each layer, and the number of neurons in the hidden layers. Therefore, for the better performance of ANNs, the only available technique is a trial-and-error algorithm.

The performance of any ANN model in anticipating the output variables can be measured and evaluated by some statistical correlations (Halali et al. (2016) [20], Govindan et al. (2018) [21]):(1)$$\mathrm{Average}\;\mathrm{correlation}\;\mathrm{factor}\;\left(R^{2}\right)=1-\frac{\sum {\left({\left(W\;\%\right)}_{est}-{\left(W\;\%\right)}_{\exp}\right)}^{2}}{\sum {\left({\left(W\;\%\right)}_{est}-\bar{{\left(W\;\%\right)}_{\exp}}\right)}^{2}}$$
(2)$$\mathrm{Root}\;\mathrm{mean}\;\mathrm{square}\;\mathrm{error}\;\left(\mathrm{RMSE}\right)=\sqrt{\frac{1}{N}\;\sum {\left({\left(W\;\%\right)}_{\mathrm{est}}-{\left(W\;\%\right)}_{\exp}\right)}^{2}\;}$$
(3)$$\mathrm{Mean}\;\mathrm{absolute}\;\mathrm{error}\;\left(\mathrm{MAE}\right)=\frac{1}{N}\sum \left|{\left(W\;\%\right)}_{est}-{\left(W\;\%\right)}_{\exp}\right|$$
(4)$$\mathrm{Mean}\;\mathrm{bias}\;\mathrm{error}\;\left(\mathrm{MBE}\right)=\frac{1}{N}\sum ({\left(W\;\%\right)}_{est}-\left(W\;\%{)}_{\exp}\right)$$
(5)$$\mathrm{Correlation}\;\mathrm{coefficient}\;\left(R\right)=\frac{\sum_{m=1}^{n}\left({\left(W\;\%\right)}_{exp,m}-\bar{{\left(W\;\%\right)}_{exp,m}}\right)\left({\left(W\;\%\right)}_{est,m}-\bar{{\left(W\;\%\right)}_{est,m}}\right)}{\sqrt{\sum_{m=1}^{n}{\left({\left(W\;\%\right)}_{exp,m}-\bar{{\left(W\;\%\right)}_{exp,m}}\right)}^{2}\sum_{m=1}^{n}{\left({\left(W\;\%\right)}_{est,m}-\bar{{\left(W\;\%\right)}_{est,m}}\right)}^{2}}}\;\;\;\;\;\;\;\;\;\;$$

In this ANN model, the weight-left percentage of a PP polymer would be the only predictive output variable. There are some advantages and disadvantages to using this ANN technique. One of the positive aspects is that an ANN can handle working with linear and nonlinear relationships and learn these relationships directly from the data without a predefined relationship. One the of disadvantages is that undertaking the fitting requires large memory and computational efforts (Bar et al. (2010) [22]).

## 3. Results and Discussion

### 3.1. Thermogravimetric Analysis of PP

The Thermogravimetric (TG) and the derivative thermogravimetric (DTG) at different heating rates of the of PP polymer pyrolysis are shown in Figure 1 and Figure 2, respectively. 

### 3.2. Pyrolysis Prediction by ANN Model

Based on 939 data sets, the feed-forward backpropagation optimization was applied to obtain the output parameter (weight-left %). In this network, only two parameters (heating rate and temperature) were considered input variables, and the weight-left percentage was considered to be the output variable during the non-isothermal experimental run. 

The collected data (939 data sets) was distributed randomly among the three selected subsets: training, validation, and test. The test set (70%: 657 data sets) set the network learning and recalculated the weights based on minimizing the error between the measured and predicted output variables. The validation set (15%: 140 data sets) was been used to check the performance of the network and instill confidence that the network was generalizing, as well as stopping the training process before any overfitting. Finally, the test set (15%: 140 data sets) tested the generalization of the network with a completely independent test of network generalization (Al-Wahaibi and Mjalli (2014) [23]). Osman and Aggour (2002) [24] mentioned that the large sets of data collected would help to achieve a high-accuracy model. Muravyev et al. (2021) [1] highlighted that the amount of data needed for the pyrolysis to get enough generalization should be higher than the total of the weight in the ANN network.

The value of regression correlation coefficient (R) in the first stage was considered to be a main criterion for judging the best network structure for evaluating the weight-left % as the output variable, using different numbers of hidden layers and numbers of neurons with a transfer function in each hidden layer.

The final and best ANN structure is shown in Figure 3 for both cases. This network has been optimized among many trials for the next-step stage calculation. It has ten neurons in the first and second hidden layers with (Logsig–Tansig), ($f\left(u\right)=logsig\left(u\right)=1/\left(1+e^{-u}\right)$,) and ($f\left(u\right)=Tansig\left(u\right)=-1+2/(1+e^{-2u}$) transfer functions, with a linear transfer function for the output layer.

The number of neurons in the hidden layer is an important parameter in deciding the efficiency of the ANN network. To avoid any underfitting (too few neurons) and any overfitting (too many neurons), one should choose the number of neurons in order to achieve a final optimum design (Al-Wahaibi and Mjalli (2014) [23], Qinghua et al. (2018) [25]). Table 1 shows the summary of the final ANN network model. The Levenberg–Marquardt algorithm was considered to be the most-utilized algorithm by previously reported investigators (Beale et al. (2018) [26]). To verify the applicability of various algorithms, the ANN program was run with more than one algorithm, such as the Scaled Conjugate Gradient and the Bayesian Regularization. By comparing the simulated results, the Levenberg–Marquardt algorithm was identified as the most applicable algorithm. Muravyev et al. (2021) [1] mentioned that there are some modern algorithms, such as the “Stochastic Gradient Descent” or “Adaptive Moment Estimation” algorithms, but most of the ANN applications are still limited to “Levenberg–Marquardt” or “Error Backpropagation” algorithms. Figure 4 confirms the results, which are close to the diagonal, indicating a good agreement between the experimental values on the *X*-axis and the ANN prediction values on the *Y*-axis at a minimum mean square error (MSE) value of 0.0087839 for the validation stage (Figure 5). 

The performance of the final ANN model in predicting the weight-left % was evaluated by the values of the statistical correlations. Table 2 shows that these statistical correlation values and their deviations are very low for the final ANN model. This indicates that the final ANN model can reasonably foresee the output within an acceptable limit of error (R ≥ 0.999). 

After obtaining the best ANN network with these 939 data, the final and best architecture (Figure 3) was simulated with twenty input data, using four for each heating rate (Table 3). In this simulation stage, fresh input data will be submitted to the network and the ANN network will provide a new, simulated output based on the approved architecture ANN (Figure 6). Figure 6, like Figure 4, shows how close the simulated ANN is to the measured output, confirming the good performance of the network. In addition, Table 4 lists the statistical parameters and shows a slightly high R value (>0.9900) but a very low MBE RMSE and MAE, respectively. Finally, Figure 7 shows the error histogram for the simulation set, which is drawn across the zero error. The error lies in a small range (−0.599 to 0.3811), which indicates the very good performance of the proposed ANN model. 

After finalizing the ANN model, it is important to establish a link or relationship between the input and the output. In general, a sensitivity analysis is used to check the effect of a small change in an input parameter on the output variable [27,28,29]. 

Table 5 shows the correlation between the input parameters with the output parameter using the “Correlation” function available in the “Data Analysis” toolbox in Excel. It can be shown that the weigh-left percentage is strongly “inversely” related to the temperature (−0.79722), while remaining low to the heating rate (0.04101). As mentioned above, the results presented in Table 6 show a strong correlation between temperature and weight-left percentage. With an increase in the pyrolysis temperature, the weight-left percentage was decreased. This finding can be demonstrated by the thermogravimetric curves elsewhere [30], which have an inverted S-shape.

The probability, or *p*-values, are used for finding the statistical significance of the input parameters, which were obtained from “Data Analysis” toolbox in Excel. This *p*-value provides an indication of whether it is significant or not. In this analysis, a significance level of 5% is used as criteria for judging the significance of the variable. Therefore, any *p*-value less than 0.05 will be considered insignificant. [31,32,33]. The results of the analysis are presented in Table 6.

The results presented in Table 5 and Table 6 evidence that the temperature is the more sensitive parameter; this is similar to the findings reported by Kempel et al. [34]. Temperature is a key that participates in providing the energy needed to volatize a unit mass of solids to volatiles (Stoliarov et al. (2008) [35]). Muravyev et al. (2021) [1] mentioned calculations in the black box, but some conclusions can be drawn by studying the importance of all input parameters. They confirmed our finding that the pyrolysis temperature has a higher importance (>30% relative importance) than the heating rate (25%). 

## 4. Conclusions

A PP polymer was studied via TGA at five sets of heating rates (5, 10, 20, 30, and 40 K min^−1^). The TG thermogram data obtained from TGA study were used in modeling an ANN to predict the weight-left percentage of the TGA. 

In this work, an ANN model was developed to analyze the thermal decomposition of a PP polymer. An ANN network of 2-10-10-1 with two hidden layers (Logsig–Tansig) was set up as the best network. It could predict the output variable (weight-left percentage) very precisely, with a regression coefficient value of (R = 1). This network has been simulated with fresh input and, again, the results were in very close agreement with the experimental values, with very high correlation coefficient of R (>0.9999). A sensitivity analysis was performed to determine the uncertainties in the relationship between the output and the input parameters. The temperature was found to be a more sensitive input than the heating rate on the weight-left percentage.

## Figures and Tables

**Figure 1 polymers-15-00494-f001:**
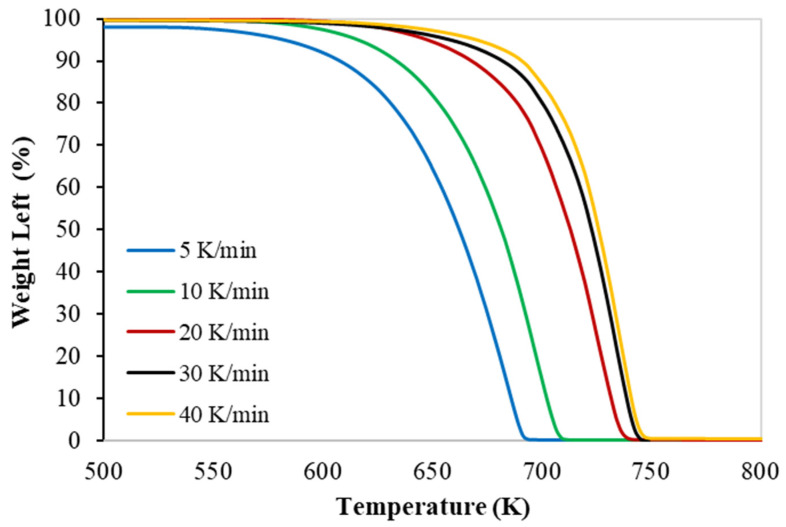
TG curves of PP at different heating rates.

**Figure 2 polymers-15-00494-f002:**
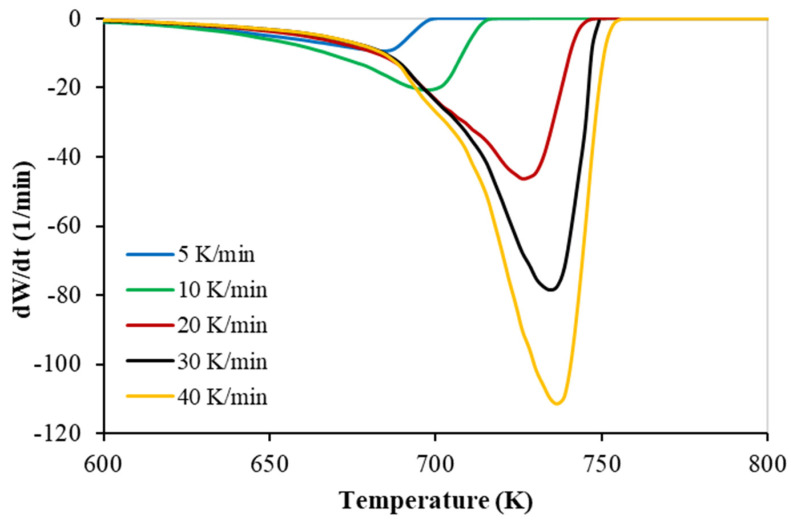
DTG curves of PP at different heating rates.

**Figure 3 polymers-15-00494-f003:**
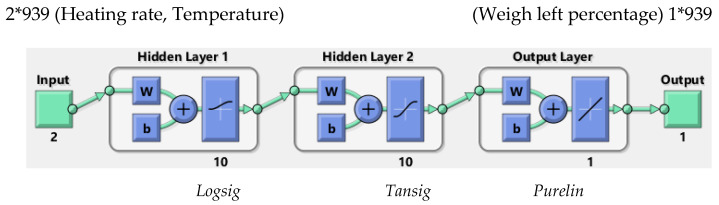
Topology of the selected ANN.

**Figure 4 polymers-15-00494-f004:**
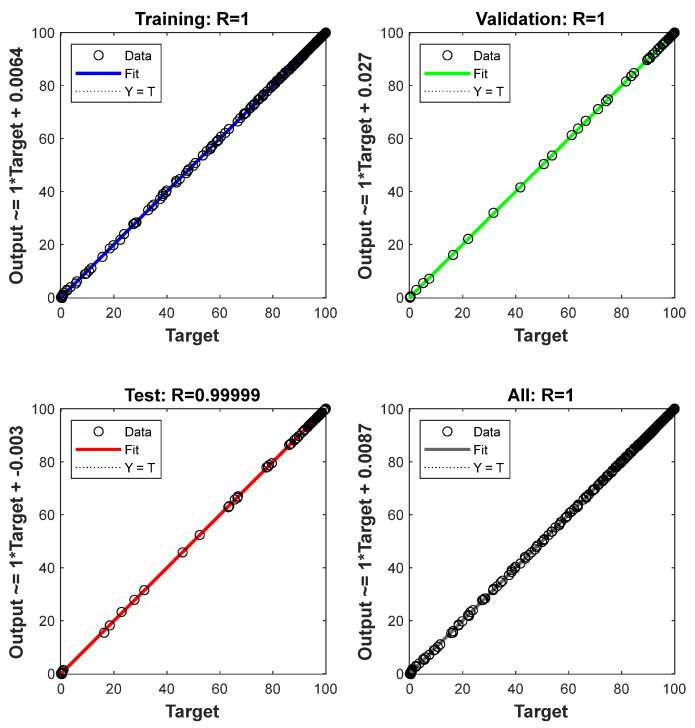
Regression of four plots for the final ANN.

**Figure 5 polymers-15-00494-f005:**
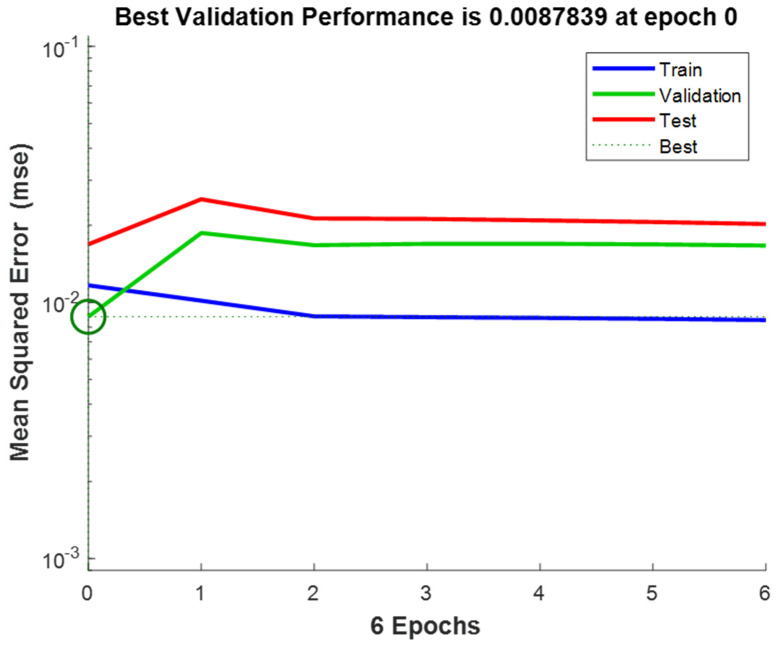
Mean square error for three plots.

**Figure 6 polymers-15-00494-f006:**
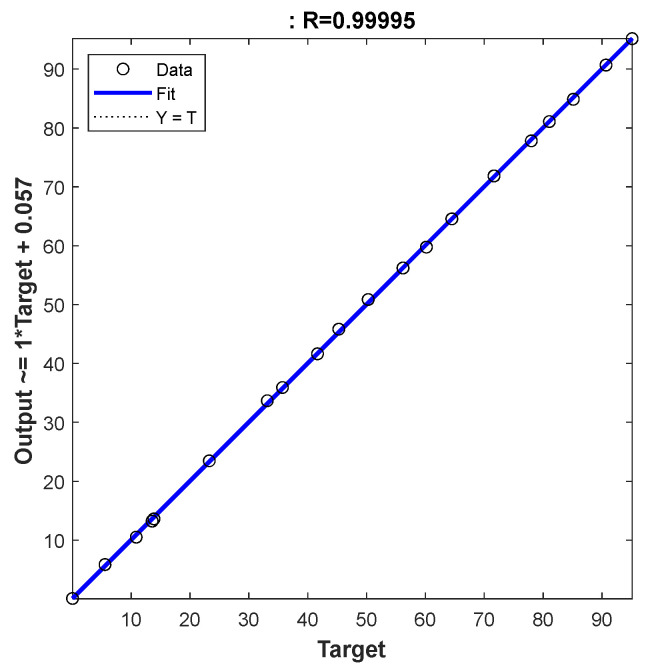
Regression of simulated data for ANN model.

**Figure 7 polymers-15-00494-f007:**
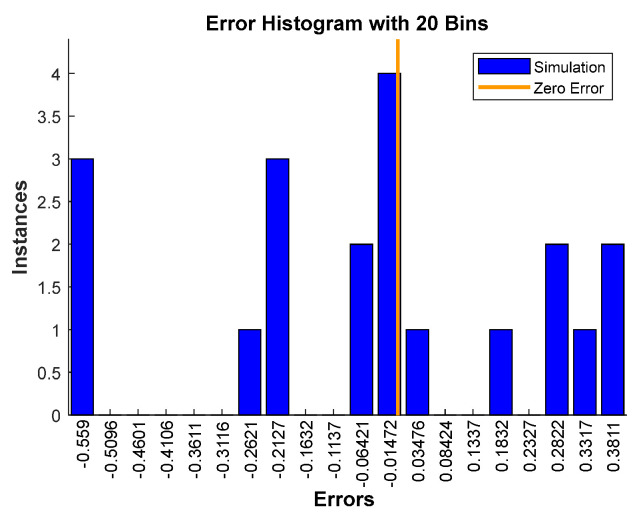
Error histogram of simulated data for ANN model.

**Table 1 polymers-15-00494-t001:** The summary of the final ANN network.

Parameters	Number
Number of inputs	2 (heating rate K min^−1^, temperature K)
Number of outputs	1 (weight left %)
Number of hidden layers	2
Network	2-10-10-1
Transfer functions (hidden layers)	Logsig–Tansig
Transfer function (output layer)	Linear
Validation checks	50
Learning algorithms	TRAINLM
Performance function	MSE
Data division function	Dividerand
Data division and number (training–validation–testing)	70%-15%-15%
	657-141-141

**Table 2 polymers-15-00494-t002:** Statistical parameters of ANN model for the training, validation, test, and all data.

Set	Statistical Parameters			
	R	RMSE	MAE	MBE
Training	1.0	0.107977	0.052428	−0.001664
Validation	1.0	0.093722	0.050369	0.000012
Test	1.0	0.129828	0.064604	−0.005936
**All**	**1.0**	**0.109579**	**0.053947**	**−0.002054**

**Table 3 polymers-15-00494-t003:** Simulation data.

No.	Input Data		Output Data
	Heating Rate (K/min)	Temperature (K)	Mass %
1	5	605.553	90.67025
2	5	668.192	41.66792
3	5	657.612	56.18587
4	5	702.357	0.05542216
5	10	651.17	81.05005
6	10	689.612	35.71684
7	10	670.339	64.48985
8	10	700.2	13.88899
9	20	698.027	71.65227
10	20	725.982	23.28277
11	20	691.698	77.98775
12	20	730.234	13.58366
13	30	717.712	60.16715
14	30	738.684	10.87517
15	30	693.819	85.1306
16	30	730.234	33.13463
17	40	725.982	50.282
18	40	742.871	5.571741
19	40	670.339	95.11706
20	40	728.123	45.28695

**Table 4 polymers-15-00494-t004:** Statistical parameters for the simulated data.

Set	Statistical Parameters			
	R	RMSE	MAE	MBE
simulated	0.99995	0.294390	0.230408	0.043524

**Table 5 polymers-15-00494-t005:** The correlation values between the input and output parameters.

	Weight-Left % (Output Parameter)
Temperature (1st input parameter)	−0.79722
Heating rate (2nd input parameter)	0.04101

**Table 6 polymers-15-00494-t006:** The *p*-values of the input parameters.

Input Parameter	Temperature	Heating Rate
*p*-value	<0.05	>0.05

## Data Availability

Not applicable.

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
