# Peer review of "Artificial Neural Network Study on the Pyrolysis of Polypropylene with a Sensitivity Analysis"

_polymers, 2023, doi:10.3390/polym15030494_

Round 1
Reviewer 1 Report
The topic can help to increase the knowledge about the utilization of ANN in the prediction of the weight left during the pyrolysis of polypropylene (PP). While, after reading the manuscript, the reviewer find that the manuscript was poorly prepared, and there are many issues that need to be polished. Some suggestions:
1. The title was not good. There was a grammar problem, pyrolysis polypropylene polymer? And “ANN” was not included.
2. Please pay attention to some typos, such as “K min-1”, “Correlation coefficient (R)”, “TGA; ;”, “weight left %”, etc.
3. Why did choose heating rate (K min-1) and temperature (K) as input variables? The reviewer means that there may be other variables that have greater impact on the output result, such as the changeable temperature rise schedule.
4. The authors mixed the two citation ways of reference, either “xxx et al. [a]” or “xxx et al. (year)”.
5. The introduction was also poorly written. The authors should summarize previous work, not just list them. And the statement needed to be closely related to the planned research work.
6. There should be more information about Figures 1 and 2.
7. In the section of 2. Materials and methods, the information of used raw materials was missing, and the research methods were not introduced in detail.
8. The text shown in Lines 123-126 was meaningless.
9. Some figures were not well displayed, such as Figures 3-7.
10. Please concise the conclusions point by point in the order of the research work.
Author Response
Response to the Comments of Reviewer 1
Comments and Suggestions for Author
Comments and Suggestions for Authors
The topic can help to increase the knowledge about the utilization of ANN in the prediction of the weight left during the pyrolysis of polypropylene (PP). While, after reading the manuscript, the reviewer find that the manuscript was poorly prepared, and there are many issues that need to be polished. Some suggestions:
- The title was not good. There was a grammar problem, pyrolysis polypropylene polymer? And “ANN” was not included.
Comment: Thank you for this comment
Action: I have modified title in my revised manuscript as suggested.
- Please pay attention to some typos, such as “K min-1”, “Correlation coefficient (R)”, “TGA; ;”, “weight left %”, etc.
Comment: Thank you for this comment
Action: I did all necessary corrections in my revised manuscript accordingly.
- Why did choose heating rate (K min-1) and temperature (K) as input variables? The reviewer means that there may be other variables that have greater impact on the output result, such as the changeable temperature rise schedule.
Comment: Thank you for this comment
Action: I agree with you if we have more experiments results taking into consideration more variable such as: catalyst ratio if you use catalyst, or particle size if I run the TGA with different particle sizes, etc. I will incorporate all your suggestion in our future work
- The authors mixed the two citation ways of reference, either “xxx et al. [a]” or “xxx et al. (year)”.
Comment: Thank you for this comment
Action: I fully agree with you the normal citation style that you have mentioned here. However, I followed here the text reference style as per Polymer journal reference format. Therefore, I am allowed to follow either to use “xxx et al. (year) [a]” if it is necessary or only “[a]” if it is at the end of the sentences.
- The introduction was also poorly written. The authors should summarize previous work, not just list them. And the statement needed to be closely related to the planned research work.
Comment: Thank you for this comment
Action: I revised the full manuscript including the introduction section and did the necessary changes as suggested.
- There should be more information about Figures 1 and 2.
Comment: Thank you for this comment
Action: More details information are available in my previous publication which mentioned in my revised manuscript.
- In the section of 2. Materials and methods, the information of used raw materials was missing, and the research methods were not introduced in detail.
Comment: Thank you for this comment
Action: More details information are available in my previous publication which mentioned in my revised manuscript.
“More details about the TGA brand and the procedure including material characteristics such as proximate and ultimate analysis data are available in my latest publication [14].”
- The text shown in Lines 123-126 was meaningless.
Comment: Thank you for this comment
Action: These statements have been rephrased to be more meaningful:
“There are some advantages and disadvantages for using this ANN technique. One of these positive aspects, ANN can handle working with the linear and nonlinear relationships; learn these relationships directly from the data without predefined relationship, while one the of disadvantages is doing the fitting needs large memory and computational efforts (Bar et al. (2010) [18]).”
- Some figures were not well displayed, such as Figures 3-7.
Comment: Thank you for this comment
Action: All these “Figures 3-7” have been improved.
- Please concise the conclusions point by point in the order of the research work.
Comment: Thank you for this comment
Action: The conclusion has been rewritten pointwise based on my research work.
Many thanks for your kind understanding.
Thanks again with best regards,
Dr. Ibrahim Dubdub

Reviewer 2 Report
The author present a Machine Learning Study of pyrolysis. The manuscript is well written, the instroduction provides all the information to really understand the paper, even for people not-in-the-field. The discussion and conclusions are very focused and well explained.
Author Response
Response to the Comments of Reviewer 2
Comments and Suggestions for Author
Comments and Suggestions for Authors
The author present a Machine Learning Study of pyrolysis. The manuscript is well written, the instroduction provides all the information to really understand the paper, even for people not-in-the-field. The discussion and conclusions are very focused and well explained.
Comment: Thank you for this comment
Many thanks for your kind understanding.
Thanks again with best regards,
Dr. Ibrahim Dubdub

Reviewer 3 Report
This research paper is interesting. Although, I have some comments:
1. English grammar needs a significant change.
2. Typos also.
3. Introduction need to be widened. Cite these papers on pyrolysis:
- 2020. Mechanical, microstructural, and thermal characterization insights of pyrolyzed carbon black from waste tires reinforced epoxy nanocomposites for coating application. Polymer Composites, 41(1), pp.338-349.
- 2019. Processing and characterization analysis of pyrolyzed oil rubber (from waste tires)‐epoxy polymer blend composite for lightweight structures and coatings applications. Polymer Engineering & Science, 59(10), pp.2041-2051.
4. Consider the voids in your ANN model as well.
Author Response
Response to the Comments of Reviewer 3
Comments and Suggestions for Author
Comments and Suggestions for Authors
This research paper is interesting. Although, I have some comments:
- English grammar needs a significant change.
Comment: Thank you for this comment
Action: The whole has been edited for English grammar.
- Typos also.
Comment: Thank you for this comment
Action: Same for Typos
- Introduction need to be widened. Cite these papers on pyrolysis:
- 2020. Mechanical, microstructural, and thermal characterization insights of pyrolyzed carbon black from waste tires reinforced epoxy nanocomposites for coating application. Polymer Composites, 41(1), pp.338-349.
- 2019. Processing and characterization analysis of pyrolyzed oil rubber (from waste tires)‐epoxy polymer blend composite for lightweight structures and coatings applications. Polymer Engineering & Science, 59(10), pp.2041-2051.
Comment: Thank you for this comment
Action: I will be very happy to add these papers but unfortunately, these papers are out of the scope of my research objectives and work. Anyway, we are doing some research about the stability of PLA in the near future and these two papers will be very useful for our work.
- Consider the voids in your ANN model as well.
Comment: Thank you for this comment
Action: I tried my best to fill any voids in the ANN presentation
Many thanks for your kind understanding.
Thanks again with best regards,
Dr. Ibrahim Dubdub

Round 2
Reviewer 1 Report
The manuscript has been well revised, and it is recommended to be accepted for publication.